# A Prototype that Fuses Virtual Reality, Robots, and Social Networks to Create a New Cyber–Physical–Social Eco-Society System for Cultural Heritage

**Louis Nisiotis [1],\*** , **Lyuba Alboul [2,3,4],\*** and **Martin Beer [1]**

1. Department of Computing, College of Business, Technology and Engineering, Sheffield Hallam University, Howard St, Sheffield S1 1WB, UK; mdb.shu@gmail.com
2. Department of Engineering and Mathematics, College of Business, Technology and Engineering, Sheffield Hallam University, Howard St, Sheffield S1 1WB, UK
3. Centre for Automation and Robotics Research, Sheffield Hallam University, Howard St, Sheffield S1 1WB, UK
4. Sheffield Robotics, Sheffield, UK
* Correspondence: l.nisiotis@shu.ac.uk (L.N.); l.alboul@shu.ac.uk (L.A.);
  Tel.: +44-114-225-6820 (L.N.); +44-114-225-5228 (L.A.)

**Abstract:** With the rapid development of technology and the increasing use of social networks, many opportunities for the design and deployment of interconnected systems arise that could enable a paradigm shift in the ways we interact with cultural heritage. The project described in this paper aims to create a new type of conceptually led environment, a kind of Cyber–Physical–Social Eco-Society (CPSeS) system that would seamlessly blend the real with virtual worlds interactively using Virtual Reality, Robots, and Social Networking technologies, engendered by humans' interactions and intentions. The project seeks to develop new methods of engaging the current generation of museum visitors, who are influenced by their exposure to modern technology such as social media, smart phones, Internet of Things, smart devices, and visual games, by providing a unique experience of exploring and interacting with real and virtual worlds simultaneously. The research envisions a system that connects visitors to events and/or objects separated either in time or in space, or both, providing social meeting points between them. To demonstrate the attributes of the proposed system, a Virtual Museum scenario has been chosen. The following pages will describe the RoboSHU: Virtual Museum prototype, its capabilities and features, and present a generic development framework that will also be applicable to other contexts and sociospatial domains.

**Keywords:** virtual reality; robots; cyber–physical–social systems; multiuser virtual environments; digital heritage; cultural heritage

## 1. Introduction

Museums are responsible for cultural heritage preservation, education, exhibition, and conservation of historical artefacts, to help visitors understand information and context regarding a topic or event that took place in history with the use of exhibits and/or multimedia interactions. However, simply displaying artefacts in glass cabinets, providing audio guides, leaflets, or written guidebooks that describe the artefact or location, are simply not enough to engage and entertain visitors. Therefore, the need to modernize exhibits' presentation methods is stressed in order to compete with the entertainment industry, overcoming the outdated principles of traditional museology [1]. The current generation of museum visitors is exposed to a wide range of entertainment technologies such as

social networking, computer and console games, films, as well as Internet of Things (IoT) devices such as wearables, connected smart phones, sensors, etc. The museum experience is now shifting from the traditional museum paradigm of 'featuring displays' and 'museum visitors,' to the 'museum experience' and the visitor as a 'museum consumer' paradigm [2]. To respond to this change, museums have been developing new methods of engaging visitors to maintain visitors flow and consumer satisfaction, as an attempt to avoid the negative financial impact and cultural implications of lower visitor numbers. Different types of technology have been explored over the years to attract, engage, and retain visitors through a wide range of mediums [3]. The MeSch project (Material Encounters with Digital Cultural Heritage) for example was set out to design, develop and deploy tools "*for the creation of tangible interactive experiences that connect the physical dimension of museums and exhibitions with relevant digital cross-media information in novel ways*" (http://www.mesch-project.eu). Computer-aided systems that allow visitors to navigate in 3D virtual heritage space reconstructions [4], web-based museums that exploit distributed web cultural resources [5], and 3D multiuser virtual worlds that replicate heritage environments [6] provide access to immersive desktop computer experiences; Augmented Reality (AR) technology, for instance, AR game-based applications to support heritage education [7], or using AR apps to generate 3D models from real world heritage exhibits [8]. Virtual Reality (VR), robots as guides and curators, sensors, smart phones and tablets, either separately or in combination, resulting in Mixed Reality (MR) and Extended Reality (XR), and photography-based systems which convert 2D to 3D images and also highly immersive Cave Automatic Virtual Environment experiences, are some examples introducing the concept of Virtual Museums [8,9].

A Virtual Museum is a "*digital spatial environment, located in the WWW or in the exhibition, which reconstructs a real place and/or acts as knowledge of a metaphor, and in which visitors can communicate, explore and modify spaces and digital or digitalised objects*" [10], p.46. Among the different technologies used in Virtual Museums, Robots and VR are increasingly utilised in order to provide opportunities for visualisation, teleoperation, and projection of the real world through virtual spaces. However, the potentials of using VR and Robots in combination with social networking technologies to allow the development of multifaceted environments that would merge real with virtual worlds dynamically are yet to be explored. This paper focuses on demonstrating the development of a Cyber–Physical–Social Eco-System (CPSeS) capable of seamlessly blending the real world with virtual social spaces by intertwining diverse technologies, including real and artificial agents and elements capable of dynamically interacting, reflecting, and influencing each other with the interactions engendered by humans and their behaviour. To demonstrate the potential of the proposed system, a Virtual Museum case study methodology has been implemented and the work in progress is discussed in the following pages. The interdisciplinary approach of the project described in this paper provides opportunities for addressing important fields in robotics, computing, and IoT societies, for instance, social computing, human–robot and human–computer interaction, human–centric physical systems, interconnected systems and agents.

## 2. Background

### 2.1. VR in Museums

VR is frequently used in museums to display, reconstruct, and/or virtually restore artefacts, locations, and archaeological sites that may have been damaged or perhaps disappeared in time [11–14]. Compared to tools used in traditional museology display practices [15], VR offers highly immersive and engaging experiences to support visitors in retrieving and adapting to information about artefacts and exhibits [16] due to the technology's affordances of immersion and presence. Presence is, "*the subjective experience of being in one place or environment, even when one is physically situated in another*" [17], p. 255. Immersion is the experience of a technology that exchanges sensory input from reality with digitally generated input [18]. Presence is similar to immersion but distinct [19], as it is the subsequent reaction to immersion leading the users' brain reacting to the virtual environment in the same way as it would have reacted to the real world [20]. The affordances of presence and immersion, combined with the

ability of VR to allow the development of environments that may be difficult or impossible to experience in the real world, are attributes that support motivations for technology adoption [21]. Despite initial adoption resistance [22], VR is now increasingly employed in museums to support and improve visitors' experiences and interaction with cultural heritage [23]. VR has, until recently been known as an expensive and resource-intensive technology challenged by technical issues and requirements. For this reason, many attempts to design cultural heritage experience in 3D virtual worlds were first taking place during the last decade using desktop virtual worlds such as Second Life and OpenSim. In these environments, visitors were interacting with the virtual world through their computer screen, keyboard and mouse. This was enabling a much more efficient and cost-effective access to immersive cultural heritage experience than VR, during a time in which these environments were trending as the next big thing in social virtual worlds' experiences. However, these environments never really met the high expectations which many of the virtual worlds' enthusiasts were hoping for, mainly due to the commercialisation of product and because virtual world designers moved on to private and personalised solutions. This resulted in users losing their interest and the technology eventually becoming outdated. Nevertheless, recent technological advancements have enabled significant technological leaps and VR is now an affordable and mature customer-ready technology [24]. It is now at a stage which can even be adapted to any smart phone, reducing the dependency on expensive hardware requirements and difficulty/inconvenience associated with the use of Head Mounted Displays (HMD) [25]. Therefore, utilising the power of smart phones that visitors have in their pockets can become mainstream in the museum experience, especially with the availability of low-cost HMD such as Google Cardboard (https://arvr.google.com/cardboard/), enabling visitors to use their smart phone device to immerse in virtual environments.

## 2.2. Robots to Support Museums

The area of robotic technologies is constantly evolving and allowing robots to become complex creations used in a wide range of domains. With the recent advancements in research, technology, and application of robotics, robots are now employed in various 'roles,' for instance, as service providers [26], assistants [27], guides [28], and used in many different fields, such as environment exploration, search and rescue, agriculture, and others. These affordances have influenced the increasing use of robots at museums to support and enhance visitors' experiences and engagement with culture and society by providing interactions between humans and robots. For the past twenty years, for instance, there has been an increasing interest in the development of robots as guides. Some early examples are Rhino [29], its successor Minerva [30], and the robots used in the Mobot Museum experiment [31]. The Tokyo Science Museum features humanoid robot guides that show visitors around [28], and the robot called 'Linda' was greeting visitors upon their arrival at the Natural History Museum in London [32]. The key features of those robots were autonomous navigation and collision avoidance systems. As the interest in robots as guides continued developing over time, more emphasis was put on the human–robot interaction. RoboX project, for example, included 11 robots, specifically to guide visitors at the Swiss National Exhibition Expo.02, using dynamic scenarios to manage visitor flow [33]. Shiomi et al. [34] have developed and placed two robots in the science museum in Osaka, responsible for engaging in personalized interaction with the museum visitors. The robot Rackham [35] was responsible for continuously giving feedback to visitors to help them understand that it knew where they were, what it did, where it went, and what its intentions were. The Robovie tour guide [36] example adopted typically human interaction cues to attract and engage visitors' attention. BeamPro by Suitable Technologies has partnered with eight US museums to provide self-guided tours to people who physically cannot pay a visit, by offering telepresence and teleoperation. BeamPro features a tablet screen and a high-quality camera, a microphone, and speakers to allow visitors to interact with each other and to view galleries and exhibits from around the world from home using a laptop and internet connection [37]. Another example involved two robots (MAVEN and EDGAR) developed for the Chinese Heritage Centre by Nanyang Technological University in Singapore, to explore the use of robotic technologies and

VR within heritage museum settings. MAVEN was a virtual human character mounted on a mobile robotic platform serving as a museum guide, while EDGAR was a humanoid robot programmed for bilingualism, adding elements of culture learning and education [38]. These robot guides worked in a single physical space together with their visitors, and they could interact in several different ways. However, the purpose of these robots and the examples presented were mainly to guide visitors to several exhibits, provide information about them, and help with routing. The main issue with these examples is that they are not helping in developing the overall story of the visit and do not provide a comprehensive social interaction experience to the visitors.

### 2.3. Cyber–Physical–Social Systems (CPSS)

The evolution and continuing advancement in research and development of technologies over the past decade influenced significant increase in the interest around complex computer systems development. A domain which drew a lot of interest is the area of Cyber–Physical Systems (CPS). CPS are computer systems employing integrated computational and physical capabilities such as sensing, communication, and actuation to physical world, with a focus on the integration of physical and information systems [39]. Such systems are commonly found in the industry for manufacturing, environmental, process and security control, aviation systems, smart structures, and many other domains [40]. However, CPS are missing the element of human interaction and social input, solely focusing on the integration of the computational with the physical world. Humans learn and interact in socially constructed ways. Hence, if CPS are used to improve and enhance the quality of human life, the creation of a "*cyber-physical society, which already includes human, social, cultural spheres as well, above the physical and cyber spaces*" [41], p.11 should then be considered. Incorporation of the human factor together with CPS is becoming a research trend and is referred to as Cyber–Physical–Social Systems (CPSS). A CPSS is a complex system constituted by a physical system, its social system including human beings, and the cyber system that connects them [42]. The CPSS is an extension of the CPS, with emphasis on the human factors influencing the system [43]; in simple words, a CPS connects the physical world to the cyber world, while a CPSS introduces the connection of the social world [44]. Figure 1 illustrates a simple yet complete CPSS where remote users can interact with the cyberspace, the physical space, and with each other through a system supporting read-and-write actions and comprised of sensors and actuators. Each user's behaviour and social interaction are being communicated back into cyberspace, enabling the system to develop an individual profile of their social characteristics and relations, rather than simply collecting information about user activities based on sensor data. Users can interact and change physical objects in the physical space using actuators, and the updated information is fed back into the cyberspace to reflect the current situation. Also, the status, interests, and knowledge of the user evolve as they navigate and engage in social interactions in cyberspace.

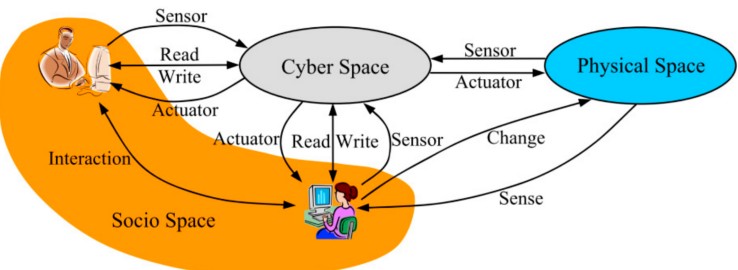

**Figure 1.** The Emerging Cyber–Physical Socio Space [45].

Over the years, several technologies have been used to support CPSS and enable bidirectional interaction between physical and cyber components such as Web 2.0 services, Video/Audio, Portable and Wearable Devices, Internet of Things, Sensors, Drones, GPS, Artificial Intelligence, Blockchain,

Robots, VR/AR and others [46]. Especially the combination of VR and Robotics has been trending in the scientific, industrial, entertainment, and enterprise communities recently, with investigations focusing on bringing together multimodal robotic sensing and computations with human–computer interfaces offering immersion and interaction in artificially created and simulated environments. This synergy of technologies establishes a new virtual reality paradigm which, "*with the help of robots, the changes made in the virtual world are 'projected' into the physical world*" [47], p.412, bridging the gap between virtual reality and robotics and offer opportunities in controlling and operating robots.

The potentials offered by VR and Robots in cultural heritage are extremely appealing and represent a challenge to experts for experimenting with new ideas and products; however, there are only a few cases of these potentials exploited in the heritage sector [48]. In addition, going beyond the topics of VR for visualisation, robot teleoperation and remote attendance, the opportunities offered by VR and Robots when combined with the power of Social Networking, are yet to be explored. Social Networking can be used not only to establish communication, but to also create mechanisms of developing individual user profiles and merging remotely with physical users. The potential of a multifaceted CPSS capable of bridging the real with the virtual and social spaces by intertwining diverse technologies will provide opportunities for exploring unique methods of complex systems development. Such a system will blend physical and virtual environments, and connect its users with real and artificial agents and elements, capable of dynamically interacting, reflecting, and influencing each other. To demonstrate the potential opportunities offered by such a system, a Virtual Museum prototype has been developed, discussed in Sections 3 and 4, respectively.

## 3. Methodology

### 3.1. Scenario-Based Approach

To investigate how new and disruptive applications can be utilised effectively, the development of a scenario can be of great benefit to enable understanding requirements and investigating potential applications, which would allow a general application to be developed. Thus, a Virtual Museum scenario has been chosen to demonstrate the attributes of the proposed system. Virtual Museums can be an example of CPSS when designed and developed in ways that would integrate the computational physical elements that interact with, reflect, and influence each other, with the systems and information exhibited by humans' and their social behaviour [49]. The fusion of VR, Robots, and Social Networking technologies can create new approaches to protect and educate people about cultural heritage, make archaeological sites and artefacts available for a wide range of the public, improve and enhance the experience and current ways in which museum visitors interact with exhibits. These affordances can contribute to the development of a new and innovative framework, a kind of CPSS that would connect museum visitors to events and/or objects, or phenomena, separated either in time or in space, or both, and provide virtual social rendezvous [48,50,51]. With the combination of these technologies in the development of the proposed system, realistic representations of archaeological locations at various moments of their timeline can be developed which, with the addition of characters/avatars to the 3D representation, a 3D+T(time) scene reconstruction can be created. This will allow visitors to explore and to see events in the past, and interact with the real and the virtual world through the use of robots as actuators. The system will provide the interface between visitors, the real and virtual worlds. Robotic agents and their corresponding virtual avatars will link the physical with the virtual and social spaces, and work together to assist remote or local visitors by responding to their personalised requests in real time.

The project described in this paper refers to such a system as a Cyber–Physical–Social Eco- System (CPSeS), that tightly integrates the physical, cyber, and social worlds based on their interactions in real time. The integration of CPSeS in a Virtual Museum provides a novel platform to enhance the enjoyment of all visitors and the issues raised by them [48,50,51]. This project envisions near future,

where the implementation of such CPSeS in a plethora of domains would lead to the development of Cyber–Physical–Social Eco-Society systems.

*3.2. The Proposed Framework*

Based on the Virtual Museum scenario, Figure 2 illustrates the basic architecture of the proposed CPSeS:

- The Users who are the Visitors of the museum(s) in this scenario are either in the Virtual World or in the Physical Space(s):

    ○ Virtual Visitors/Users can navigate in the environment which can range from a realistic representation of the Physical Space, to an environment that deviates from reality.
    ○ Real Visitors may visit the Physical Space(s), navigate, observe, and interact with the artefacts and exhibits, learn about their history, etc.

- The Integration Layer is responsible for:

    ○ Establishing the connection between the Virtual World and the Physical World through Robotic Agent(s) placed in the Physical Space(s). The Physical Space(s) will have robotic infrastructure installed to support this.

- Robotic Agents are responsible for:

    ○ Actuation in the Physical World based on actions requested through the Virtual World and handled through the Integration Layer;
    ○ Guiding Physical and Virtual Visitors;
    ○ Providing Video and Audio feed to the Virtual World through the Integration Layer.

- The Communication Layer is responsible for:

    ○ Establishing communication between the Remote Users and Virtual Guides in the Virtual World;
    ○ Utilising the Robotic Agent(s) to remotely communicate with Physical Visitors in the Physical Space(s);
    ○ Supporting the capability of Remote-to-Physical Visitors' communication in the form of an Online Post-it Wall that will link the Remote with the Physical Visitors directly through the use of Tablet devices placed on the Robotic Agents in the Physical Space, and through Virtual Walls in the Virtual World.

        ▪ Users can post notes and messages through the Virtual World, and Physical Visitors can respond to them through the Tablet interface in the real world in sync.

An ambition for a large-scale application of this framework envisions a system capable of connecting remote and physical museum visitors to events, locations, or phenomena that are separated either in time, or in space, or both, and allow them to experience an immersive combination of real and virtual spaces in social ways through the use of VR, multilinked robotic agents, and avatars.

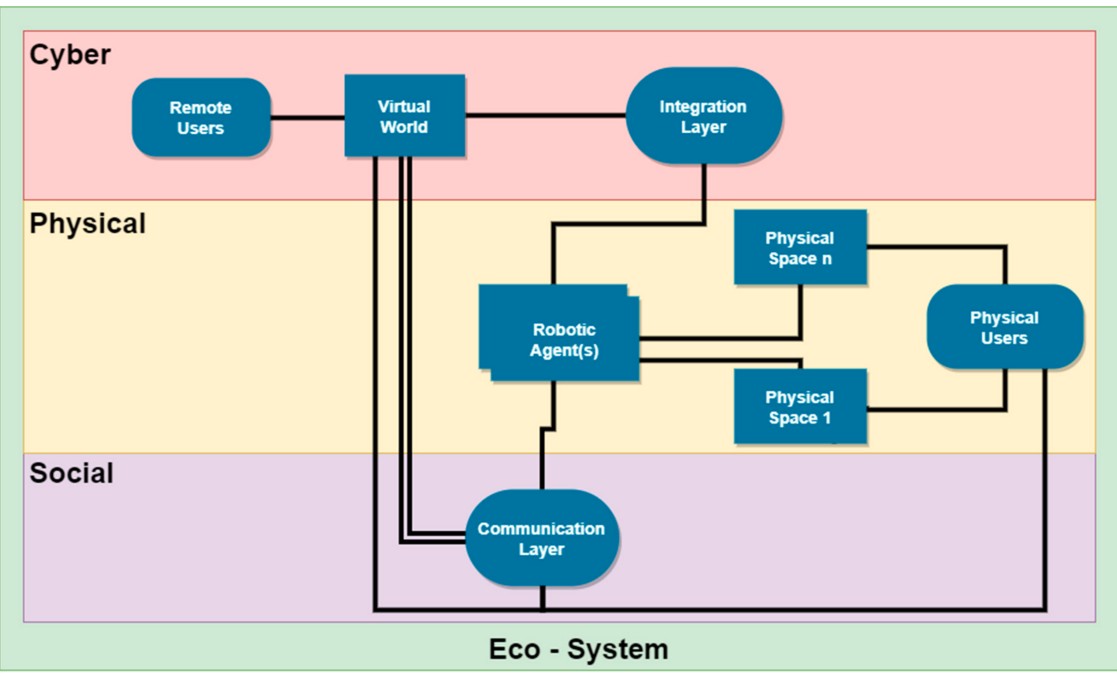

**Figure 2.** The Proposed Cyber–Physical–Social Eco-System (CPSeS) Framework.

## 4. The Virtual Museum Prototype

To explore the possibilities of the proposed CPSeS and initiate the implementation of the proposed framework, an initial demonstrator has been developed with which to conduct experiments. The RoboSHU prototype is the Virtual Museum of the History of Robotics, experienced in VR using smart phones and low-cost HMDs for now. The RoboSHU is designed to provide information about the history of robotics and promote the work of the Centre for Automation and Robotics Research at Sheffield Hallam University, feature and promote students' work, and exhibit virtual robotics artefacts. The main concept of RoboSHU is to provide Cyber–Physical–Social functionalities that would support the development of a networking environment that brings together the real with the virtual space, through the influence of its users and their behaviour.

To provide Cyber–Physical functionalities to the system, the RoboSHU features a real robot (Fetch54), which is connected to its virtual robotic avatar in the virtual environment, acting as an intelligent agent that connects the real with the virtual world (see Figure 3).

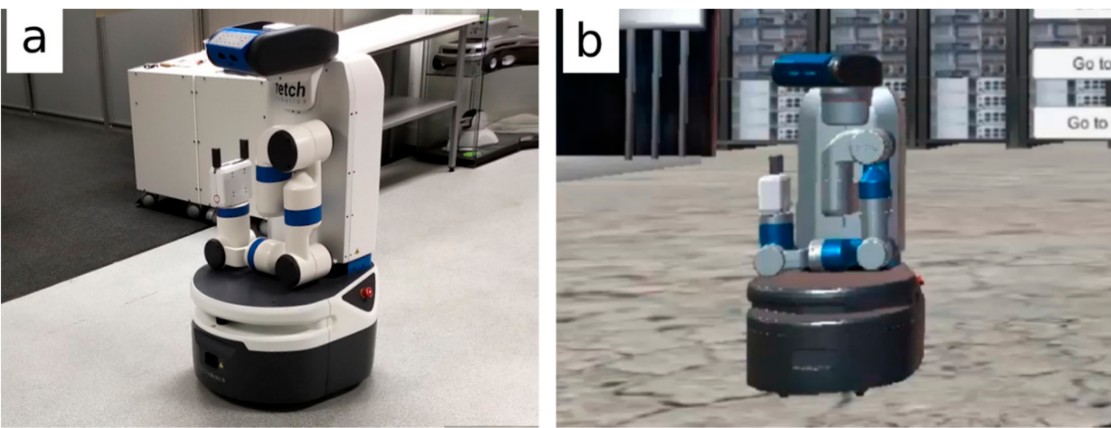

**Figure 3.** The Real (**a**) and the Virtual Robot (**b**).

Fetch54 is a mobile manipulation robot from Fetch Robotics [52] designed to be the standard platform for the next generation of mobile manipulator applications [53]. It provides a high-resolution camera, laser scanning, multiple sensors suitable for perceiving objects, navigation, and manipulation in dynamic environments, and a mechanical arm designed to be able to reach and grab items. Fetch is placed in one of the exhibition areas of the host University. The area has been scanned and mapped using Fetch's laser scanning technology, and a C++ application to identify predefined locations in the real space was developed, where Fetch should navigate when instructed by users through the virtual environment. The Virtual Fetch is synchronised with the real Fetch's movements and actions and provides live video feed of the real world. An example is depicted in Figure 4. The image on the left features a view of the real world where Fetch robot is looking at the different robots and other equipment that are placed in its front. The right side of the figure shows a screenshot of the virtual environment where the Fetch digital twin can be seen, together with a plane object attached to its front that shows the camera feed directly from the eyes of the robot. When Fetch is moving in the real world, then its digital twin is synchronously moving and replicating its movement in the virtual environment.

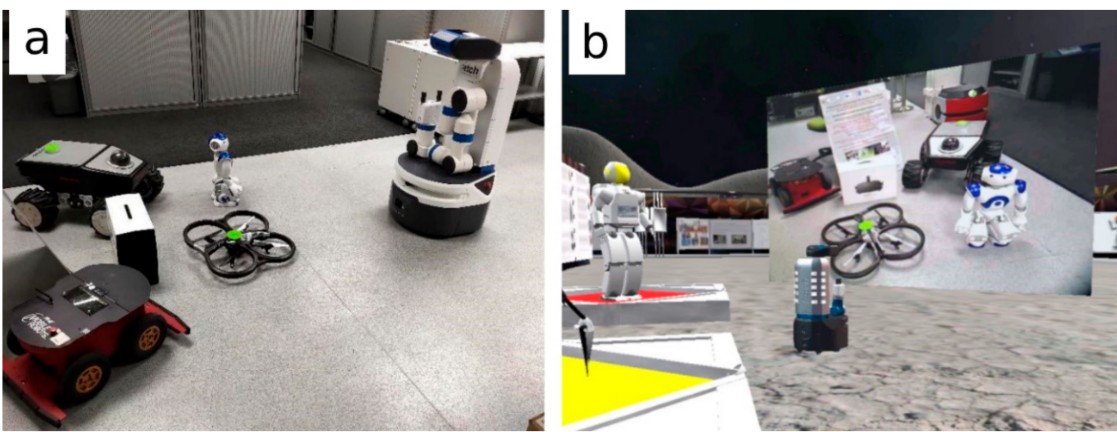

**Figure 4.** The view of the real world (**a**) through the eyes of the robot in the virtual environment (**b**).

To enable user interaction in the virtual world while users are experiencing the environment through their HMD, the single-click functionality provided in standard cardboard-based HMD like Google Cardboard is utilised. When users click and hold the HMD button, their avatar is moving forward, and when double-clicking, it is moving backwards. To enable interaction with elements within the virtual world, the VR interaction system is handled though a reticle pointer which expands when the user is focussing on an element that has interactive behavior attached to it. Through this way, users can use different buttons attached on the robot, sending system requests on where to navigate them using the RosBridge technology explained in Section 4.1.3 below. Users will be able to interact with several artefacts that exist in the virtual world, where, by clicking on them, they will be able to find out more information about the exhibit, or to preview them by starting animations demonstrating how the robot works in reality through the virtual world.

To support the Social component of the proposed CPSeS, networking capabilities to the RoboSHU have been implemented to provide Multiuser capabilities to the environment, which, however, is still at its early stages of development. The environment allows 20 multiple users to connect and coexist at the same space and at the same time, and see each other's avatars as standard robot shapes. Communication, avatar registration, and other functionalities are planned to be designed in the future. The environment layout to date has many informational spaces and is designed in ways to help visitors gather information and to interact with objects in the virtual space. Social spaces and additional areas are yet to be designed in more detail in the future.

*4.1. Technology*

4.1.1. Unity 3D

To develop the prototype, Unity 3D game development platform is used. Unity 3D [54] is a cross-platform and open-source all-in-one environment allowing to develop and modify high-fidelity graphical environments and program behaviours to objects and components. Unity features powerful rendering, physics and collision detection engines, supporting implementation and modelling of sensors, and provides support for deployment on more than 15 platforms, including Oculus Rift, HTC Vive, and Google VR platforms. Unity supports C#, which is one of the industry-standard languages; it features an asset store and is supported by a large community of developers [54].

The prototype to date is targeting the Android platform using the Android SDK and the Google VR SDK for Unity and Android. To experience the environment, visitors are using low-cost HMDs, such as Google Cardboard or similar.

To support the Multiuser component of the environment, the Photon Engine [55] has been implemented. This is a cross-platform Software-as-Service solution that provides synchronous and asynchronous capabilities and is used widely in the game development community to support synchronous networking. The free version of Photon Engine has been adopted, supporting up to 20 concurrent users to connect to the environment, but it can be scaled to support a higher number of concurrent users based on demand for a subscription fee.

4.1.2. ROS

To develop the functionalities of the Robot, the Robot Operating System (ROS) framework has been used. Over the years, many middleware systems have been developed to provide common interfaces to control services, code sharing, and reusability to support robotics deployment and development [56]. These systems provide communication mechanisms, application interface for coding, libraries, and drivers. Their infrastructure usually includes messaging and protocol support between different nodes connected to robotic hardware. One of the most used middleware is the Robot Operating System (ROS), introduced in 2007 by Willow Garage [57]. ROS is a peer-to-peer, tools-based, multilingual, free, and Open-Source framework aiming to address the difficulties in writing software for robots, as the scope of robotics is growing and the need of standardisation and reusability is imperative [58]. Compared to other Robot middleware systems, it is one of the largest efforts to foster a robotics community to support code sharing and building on each other's work. ROS provides multinode and independent node support coming from a pre-built system, as well as several systems, tools, and algorithms to provide the basis for controlling complex robots [56]. Some of these tools are programming languages such as C++ and Python, to program and connect to interfaces of the robots, graphical user interfaces (GUI) such as RViz for visualisation of robot sensor data, and many others [59].

ROS provides a set of tools and libraries based on a publish/subscribe architecture service distributing data between different nodes. A master service is responsible for managing subscription and publication of topics. These topics are essentially information channels in which data from different nodes are streaming from and to different machines. Nodes are processes that perform computation and communicate with each other by passing messages. Nodes publish ROS messages of several topics and/or create subscribers to receive and manage incoming publication on those topics [60]. Multiple nodes are capable of concurrently publishing and/or subscribing to the same topic; as well, a single node is capable of publishing and/or subscribing to multiple topics.

4.1.3. RosBridge and ROS#

To facilitate connectivity between Unity and ROS, and therefore to the robot, the RosBridge framework and the ROS# (ROS Sharp) [61] Unity plugin has been implemented. RosBridge is an additional abstraction layer operating on top of ROS, which provides a simple web socket-based

approach to enable programmatic access to ROS and robot interfaces [56]. To establish communication and connect RosBridge with Unity, the Siemens ROS# Unity plugin developed by Martin Bischoff is utilised [61]. This is a set of open-source tools and libraries tool comprising of the RosBridgeClient, which allows connecting to RosBridge, and the UrdfImporter tool to allow importing and exporting robot information.

In Unity, a WebSocket client component developed in ROS# is implemented to interface with RosBridge. Through this client, the prototype is publishing and subscribing to different topics on the robot, and RosBridge is handling the connection and information parsing from the game engine to the robot and vice versa. Using this infrastructure, the virtual environment is subscribing to topics published by Fetch robot, for instance, video feed, movement, and other data, and information is then fed back to the virtual environment, providing a Cyber–Physical synchronisation.

### 4.1.4. Initial Functionalities of the Virtual Museum Prototype

The prototype to date is at its initial development stages and features only a limited subset of the intended functionalities.

To set up the experience, the robot must first be placed in the exhibition area, and connected to the network. Users also have to download and install the app on their Android smart phone device. After placing their smart phone into the HMD, users are able to interact with the welcome screen, which asks them to choose between a single or a multiuser experience. Once the user chooses the experience required, a connection with the robot is established, and the real robot's digital twin is generated in the virtual environment. Users can navigate in the virtual world and study the information provided through several information boards and videos, and interact with some of the robotic exhibits, each featuring a different set of functionalities.

The users can see the real world through a video feed object which is attached to the front of the virtual robot. On the side of the robot, a number of buttons provide different instructions to the robot. For instance, one of the buttons sends the robot to the Helpdesk in the real world. Other buttons send the robot in front of a particular robotic exhibit placed in the exhibition area, informational boards, and other areas in the real world. Once a user chooses a location, the system submits a request to the real robot, which navigates towards it, and its movement is replicated by its digital twin. Once the robot completes the instruction, users can then choose the next location.

This initial prototype is a proof of concept, and a more detailed implementation of the proposed framework described in Section 3.2 is underway.

## 5. Discussion

Recent rapid advances in mobile cloud computing, emerging forms of digital communications, such as man–machine, machine–machine communications, pose fundamental questions and challenges about the role of modern museums and cultural heritage sites in their mission to collect, preserve, exhibit, and interpret works of art, "*addressing the relationship that connects the artwork, museology and digital technology has become a key dialog within cultural practice and policy*" [62], p. 51. More and more museums made transitions from static displays and traditional interpretation of artefacts to new artistic practices that have shifted from static to dynamic. One of the epitomes of this new shift is interactive, Digital Art Museum: teamLab Borderless (https://borderless.teamlab.art/), opened in Tokyo in 2018. Its creators claim that it is the world's largest museum dedicated to digital, interactive art [63]. Our system incorporates this new dynamic, interactive, and temporal art of exhibition, but goes further; by integrating cyber–physical systems and social networks, it provides a novel platform with which to address the challenges in Cyber–Physical–Social interactions and human–centric technology development. The system described in this paper aims at providing much richer functionality than traditional CPS by seamlessly blending the real with the virtual worlds and transforming the way humans interact with each other and with nonhuman resources. Physical robots, together with their virtual avatars and other virtual and physical objects of interest, would form an integrated and

intelligent multiagent system capable of interacting with humans and be influenced by their behaviour. Through this system, visitors can access immersive, social, and personalized spaces, where they can interact with objects, characters, and with each other to better understand the meaning of the spaces that they are visiting. This system can provide accessible immersive experiences and can enable to bring together exhibits that are displayed elsewhere, or have been removed for conservation, are on loan, may have been lost through fire, conflict, etc. Both local and remote visitors will be able to appreciate the exhibits in a variety of settings and forms, both physical and virtual, where items and their associated information can be shown in the virtual world, etc. The ability of delivering content through such a system will result in a much more interactive, sociable, and immersive delivery method, which will enable better understanding, create memorable experiences, and increase cultural awareness. The development of this system can assist in globalisation, availability, accessibility to, and provide immersive telepresence in museums' premises and socio-immersive personalized visits.

With the development of the proposed system, new approaches can be created, not only to protect cultural heritage sites and make them available for a wide range of public, but also create a generic CPSeS framework that will allow for connecting users to events separated either in time or in space, or both, and facilitate virtual rendezvous between them. The successful development of the proposed CPSeS will provide an effective method of developing highly interactive systems that allow humans and autonomous systems to simultaneously operate and interact in the real and virtual spaces.

Moreover, the proposed concept can be perceived in a much broader sense, as a new type of Cyber–Physical–Social Eco-Society systems that can be applied to a plethora of domains, which could have significant economic and societal impacts upon them, such as education, warehousing, auctioning, smart cities, climate control, disaster prediction, environment monitoring, and others. Consequently, the term visitor can be seen in a broader sense, ranging from a museum visitor, a tourist, a customer, a museum curator, an operator, a student, a shop assistant, an auctioneer, etc.

Future work is under way to provide additional functionalities to the system. The research team is currently developing a registration and avatar selection system to allow users to create their profiles, customise their selected avatar shape and appearance. The team is also planning to involve additional robots and connect them in the virtual world to trial and understand their capabilities and limitations. Immediate future work involves further development and implementation of robotic and environment behaviour. An example is the development of specific functionalities to allow the robot to act as a CPS Guide that would meet and greet visitors in the real world, and its actions to reflect on the virtual world. Functionalities will also be implemented to support communication between visitors in the virtual and the real world and with the robotic agents. The research team is also developing game scenarios and elements to add into the virtual museum, allowing users to participate in multiuser activities that would immerse them into a collaborative storytelling adventures.

To date, the system has been demonstrated and initially evaluated in several trials by students, colleagues, and by people from the public in order to get informal feedback to help the design and development process. Users were asked to navigate the virtual museum and interact with the environment and the robot, and were then informally enquired about their experience, their usability perceptions, and to discuss general development ideas. The prototype was initially demonstrated to the public during the 'Fictional Human and Real Robot: Sharing Spaces with Robots' series of events that took place in 2016 as part of the UK-wide Being Human Festival (https://www.shu.ac.uk/research/specialisms/materials-and-engineering-research-institute/news/being-human-festival-2016-fictional-human-and-real-robot).

When the system is developed more coherently and the intended functionalities are implemented, a series of empirical evaluation studies of the environment's efficacy, usability, ease of use and usefulness, and other environment attributes will be conducted. Human robot interaction evaluation will also take place to ascertain the efficacy of the real and virtual robotic agents' interactions with the remote and local users.

## 6. Conclusions

In our days where advanced smart technologies and the increasing use of connected things have a huge impact in life, the utilization of this fusion of technologies can enable to develop the next generation of CPS and networks. The work described in this paper can lead to emerging concepts, approaches, and to new subfields of research to shift the current ways of developing complex systems, the way we learn, experience, and understand concepts, as well as supporting operations and services.

The CPSeS discussed in this paper aims to increase interaction between visitors and objects by using a fusion of cutting-edge technologies, digital libraries, and the visitors' interactions. Such a system will allow visitors to have a much clearer impression of the importance of the museum artefacts, going beyond simply using current AR, VR technology to improve displaying of information and Robots as guides and telepresence devices. A CPSeS Virtual Museum would be able to bring related artefacts which are often spread over a number of often disparate sites, for example, and link them in a social virtual environment. The development of this system can assist availability and accessibility to cultural heritage exhibits and provide immersive telepresence in distant museums' premises and to experience socio-immersive personalized visits. The Virtual Museum scenario chosen as the case study can help to demonstrate the transformative and timely qualities of the system. The case study provides an ideal opportunity to investigate and stimulate new lines of research in many areas, leading to reinterpretation of cultural heritage towards a new shared culture around the world and to move the state of the art forward in a meaningful and useful way.

Introducing such CPSeS for cultural heritage will provide a novel platform for innovative applications and services that can benefit stakeholders (i.e., museums, government, visitors) and society in general. Such a system can move several cultural institutions away from isolation and social exclusion, widen accessibility, and provide innovative and democratic approaches to experience cultural heritage, and also help in educating society. Potentially, the adoption of such a system may encourage society to respond more positively to efforts of broadening access and to support similar initiatives.

The CPSeS presented in this paper aims at creating new approaches, not only to protect cultural heritage sites and make them available to the public, but also create a generic CPSeS framework that can be used in a plethora of domains and applications and provide an effective method of developing highly interactive systems that allow humans and autonomous systems to simultaneously operate and interact together in real and virtual spaces.

**Author Contributions:** The design of this research is equally shared between the authors. All three authors contributed towards conceptualisation, methodology, investigation. L.N. and L.A. contributed towards the development of the software and resources, and wrote the original draft. M.B. contributed significantly in reviewing and editing the draft manuscript. All authors have read and agreed to the published version of the manuscript.

**Funding:** This research received no external funding.

**Acknowledgments:** The research team would like to express thanks to the intern students from ISEN (Higher Institute for Electronics and Digital Training), Lille, France: Robin Ghys, Jean-Alexis Hermel and Léo Dedeine for their contribution to the development of the Virtual Museum, to Jacques Penders, the Head of the Centre for Automation and Robotics Research, for providing the working environment and equipment, to Andrew Alderson, the Director of the Materials and Engineering Research Institute, and Musa Mihsein, the Head of the Department of Computing for their support on this project. The authors also would like to thank the colleagues from C3RI (Cultural, Communication and Computing Research Institute), in particular, David Cotterrell and Luigina Ciolfi, for useful discussions.

**Conflicts of Interest:** The authors declare no conflict of interest.

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
