# Peer review of "A Prototype that Fuses Virtual Reality, Robots, and Social Networks to Create a New Cyber–Physical–Social Eco-Society System for Cultural Heritage"

_sustainability, doi:10.3390/su12020645_

Round 1

Reviewer 1 Report

The manuscript deals with a proposed Cyber-Physical-Social Eco-System.

In spite of the fact that the manuscript refers to the main important references and the authors have written the Introduction and Background sections clearly; some details are missing. For example, neither the validation test of the prosed prototype nor the user evaluation described. Without those tests, the paper is not finished. Please write a section about these tests.

To sum it up, the manuscript is innovative and obviously it will be interesting for readers. The reviewer suggests this manuscript for rewriting.

Author Response

Dear Reviewer. 

Thank you very much for taking the time to review and provide suggestions on how to improve our paper. Your feedback is invaluable. 

Comment: In spite of the fact that the manuscript refers to the main important references and the authors have written the Introduction and Background sections clearly; some details are missing. For example, neither the validation test of the prosed prototype nor the user evaluation described. Without those tests, the paper is not finished. Please write a section about these tests. 

Reponse: To address this comment, we are presenting information in the Discussion section about several trials we have conducted using the environment to informally obtain user opinions about the system usability and initial acceptability of the approach. In particular, we have demonstrated and enabled users to experience the environment at different stages of its development and then informally queried their opinion. We are in the process of designing appropriate experimental studies to evaluate user experience of the environment in the near future, once the environment is further developed to implement the majority of the intended functionalities.  

Reviewer 2 Report

The idea is very interesting. Introduction and background very well explain the motivation and related work. 

Can multiple users use the system at the same time, if yes how meny? What are the limitations on that?

Have you tested the presented system with users and if not do you plan it?

How many robots can be included to the system at the same time?

How users will interact with robots?

How are robots included with the cultural heritage? It is not very clear from the paper where robots are placed, so please comment on that and try to integrate more and show how robots, users and cultural heritage are connected in the real prototype of the system.

Author Response

Dear Reviewer. 

Thank you very much for taking the time to review and provide suggestions on how to improve our paper. Your feedback is invaluable. Below is our response to your comments: 

Comment 1: Can multiple users use the system at the same time, if yes how many? What are the limitations on that?  

We are addressing this now at the end of section 4.1.1. 

Comment 2: Have you tested the presented system with users and if not do you plan it?  

We have addressed this in the Discussion section, presenting information of how we have informally tested the environment with real users to date, and what are the plans for future work. 

Comment 3: How many robots can be included to the system at the same time?  

Our respond to this comment can be seen in the Discussion: The team is planning to involve additional robots and connect them in the virtual world and test and understand their capabilities and limitations. At the moment, we have only tried one real and one virtual robotic agent simulated from a computer. 

Comment 4: How users will interact with robots?  

We have addressed this comment in Section 4 by elaborating on the user interaction system, how users are interacting with the robot and with the artefacts in the environment. 

Comment 5: How are robots included with the cultural heritage? It is not very clear from the paper where robots are placed, so please comment on that and try to integrate more and show how robots, users and cultural heritage are connected in the real prototype of the system.  

To address this comment, we have introduced Section 4.1.4 providing information of the initial functionalities the prototype, describing the procedures on how users can connect to the environment, the role of the robot, its initial functionalities, and how users can interact with the robot and the environment for now. 

Reviewer 3 Report

Overall an intriguing research project about new possibilities for museums and heritage to interpret and communicate content by merging various digital technologies in physical and virtual reality. The study emphasises the importance of the social aspect of accessing heritage.

The paper is well written, I may only suggest mentioning MR and XR as "mixed" or "merged" concepts, past examples in relation to social media and virtual world and why they were relatively unsuccessful in a way we do not use them widely (anymore) - heritage/museum virtual worlds based on Open Wonderland and SecondLife for example.

The two images (Fig 3) should be better explained. It is also not clear to "non-robots" experts, which robot is a museum artefact and which one is a part of the study. "Traditionally" looking artefacts such as ancient vases and sculptures, may be helpful in avoiding confusion in the future, although it is great to present the history of robots.

Last but not least, it may be worthwhile looking into museological studies of new ways of museum communication moving away from static museum expositions. Both virtual and physical exhibitions in Fig 3 represent a relatively traditional and static museum display.

Author Response

Dear Reviewer. 

Thank you very much for taking the time to review and provide suggestions on how to improve our paper. Your feedback is invaluable. Below is our response to your comments: 

Comment 1: The paper is well written, I may only suggest mentioning MR and XR as "mixed" or "merged" concepts, past examples in relation to social media and virtual world and why they were relatively unsuccessful in a way we do not use them widely (anymore) - heritage/museum virtual worlds based on Open Wonderland and SecondLife for example.  

We have addressed this comment by mentioning MR and XR modes and introducing a brief history of the use of desktop virtual worlds such as Second Life and Opensim to support cultural heritage, and how these environments never reached the high expectations many of their enthusiasts had. 

Comment 2: The two images (Fig 3) should be better explained. It is also not clear to "non-robots" experts, which robot is a museum artefact and which one is a part of the study. "Traditionally" looking artefacts such as ancient vases and sculptures, may be helpful in avoiding confusion in the future, although it is great to present the history of robots. 

To address this comment, we have added an additional figure (Figure 3) to first show the real and virtual robot so the reader can clearly understand them and the environment they operate. We are then providing elaborated Figure explanation to help the reader understand the example shown in Figure 4 (previously 3).  

Comment 3: Last but not least, it may be worthwhile looking into museological studies of new ways of museum communication moving away from static museum expositions.  

We have addressed this comment in the Discussion section. 

Comment 4: Both virtual and physical exhibitions in Fig 3 represent a relatively traditional and static museum display.  

We have attempted to better explain the story through adding additional images and the elaborating paragraph discussed in the previous comment above. 

Round 2

Reviewer 1 Report

Thank you for considering my suggestions and you have inserted new parts into your manuscript.

Finally, may I ask you to modify the title of the paper including the "prototype" word? I recommend it because your system is in a prototype version and you are in the process of designing appropriate experimental studies to evaluate the user experience of the environment in the near future.

Author Response

Dear Reviewer. Considering your suggestion, we have decided to change the title of our paper to: A prototype that Fuses Virtual Reality, Robots and Social Networks to create a New Cyber-Physical-Social Eco-Society System for Cultural Heritage.

Thanks for your valuable feedback.